# Microstructure and Wear Characterization of the Fe-Mo-B-C—Based Hardfacing Alloys Deposited by Flux-Cored Arc Welding

**DOI:** 10.3390/ma15145074

**Published:** 2022-07-21

**Authors:** Michał Bembenek, Pavlo Prysyazhnyuk, Thaer Shihab, Ryszard Machnik, Olexandr Ivanov, Liubomyr Ropyak

**Affiliations:** 1Department of Manufacturing Systems, Faculty of Mechanical Engineering and Robotics, AGH University of Science and Technology, 30-059 Krakow, Poland; machnik@agh.edu.pl; 2Department of Welding, Ivano-Frankivsk National Technical University of Oil and Gas, 076019 Ivano-Frankivsk, Ukraine; pavlo1752010@gmail.com (P.P.); thaer.a@albayan.edu.iq (T.S.); 3Medical Instruments Techniques Engineering Department, Technical College of Engineering, Al-Bayan University, Baghdad 10070, Iraq; 4Professional College of Electronic Devices, Ivano-Frankivsk National Technical University of Oil and Gas, 076006 Ivano-Frankivsk, Ukraine; o.ivanov3001@gmail.com; 5Department of Computerized Engineering, Ivano-Frankivsk National Technical University of Oil and Gas, 076019 Ivano-Frankivsk, Ukraine; l_ropjak@ukr.net

**Keywords:** hardfacing, powder electrodes, FCAW, coating, abrasion wear, hardness, carbides, borides, chromium

## Abstract

An analysis of common reinforcement methods of machine parts and theoretical bases for the selection of their chemical composition were carried out. Prospects for using flux-cored arc welding (FCAW) to restore and increase the wear resistance of machine parts in industries such as metallurgy, agricultural, wood processing, and oil industry were presented. It is noted that conventional series electrodes made of tungsten carbide are expensive, which limits their widespread use in some industries. The scope of this work includes the development of the chemical composition of tungsten-free hardfacing alloys based on the Fe-Mo-B-C system and hardfacing technology and the investigation of the microstructure and the mechanical properties of the developed hardfacing alloys. The composition of the hardfacing alloys was developed by extending the Fe-Mo-B-C system with Ti and Mn. The determination of wear resistance under abrasion and impact-abrasion wear test conditions and the hardness measurement by means of indentation and SEM analysis of the microstructures was completed. The results obtained show that the use of pure metal powders as starting components for electrodes based on the Fe-Mo-B-C system leads to the formation of a wear-resistant phase Fe(Mo,B)_2_ during FCAW. The addition of Ti and Mn results in a significant increase in abrasion and impact-abrasion wear resistance by 1.2 and 1.3 times, respectively.

## 1. Introduction

Equipment used in mining, oil and gas, metallurgy, and other industries operate in conditions that cause severe wear from friction, aggressive environments with a high content of abrasive particles, high temperatures, and cyclic and dynamic loads. Therefore, it seems necessary to use new, advanced, economically viable technologies that make it possible to achieve high reliability of modern equipment wear parts [1,2,3,4,5,6]. A number of requirements are placed on the technical condition of mechanical engineering products. The most important among them are the reliability and maintenance of the operational properties of the product throughout its life cycle [7], since the occurrence of wear and corrosion emergencies, as well as the mechanical destruction of equipment parts, can lead to environmental risks [8].

An extension of the service life of the equipment is achieved by reducing the wear and tear of the responsible parts through constructive, technological, and operational measures. When designing devices and machines, special attention is paid to the rational selection of construction materials [9,10,11,12], the study of wear processes [13,14,15,16,17], corrosion [18,19] and tribocorrosion [20,21,22], the investigation of temperature [23,24], and the stress state of components [25,26], including those with functional coatings [27,28]. The successful healing of cracks in solids is facilitated by welded [29] and laminated [30,31,32] layers. Also important is the problem of contact interaction of coatings with monolithic abrasives and loose abrasive particles [33].

The specified accuracy and quality of the working surfaces of the parts is achieved using technological methods of mechanical processing [34,35,36], taking into account the technological heredity [37] and the rational selection of metalworking machines [38], since the occurrence of errors in the processing of parts and the performance of assembly processes leads to growth stresses in components with coatings [39].

The favorable combination of properties between the reinforced surface layer and the base metal can be achieved by using various surface engineering processes. Extension of the service life of critical parts is provided by the methods of hardening using surface deformation [40,41], electrochemical Cr plating [42,43], chemical-thermal treatment [44,45], electrospark doping [46,47], laser doping [48,49,50], laser cladding [51], plasma hardening [52,53], thermal spraying [54,55], supersonic spraying [56,57,58], vacuum-arc deposition [59,60,61], formation of oxide and nitride coatings by combined methods [62,63,64,65,66,67], manual arc hardfacing [68,69], and flux-cored arc welding [70,71,72,73,74,75,76].

A comparison of different surface reinforcement techniques shows that cored wire arc welding has many advantages, the most important of which are high productivity and coating thickness, the possibility of depositing high-alloy alloys, and ease of use. Such a method can also be used to apply hardfacing to new wear parts as well as to restore worn ones. In addition to experimental methods for the development of materials for wear-resistant coatings, the results of thermodynamic modeling [77,78], the calculation of crystal lattice properties [79,80,81], the determination of factors influencing corrosion resistance [82], and a rational choice of deposition process parameters [83,84] are very important for achieving the maximum result of surface reinforcement. When developing technological processes for depositing coatings [85] and modeling metal melting processes [86], it is also necessary to take into account the influence of external factors, as well as to apply intelligent methods for comparing energy processes [87].

As an alternative for the more expensive materials, such as W, Co, Nb, systems based on Fe-Cr-C, Fe-Mn-C-B, and Fe-Ti-C [88,89] are used more often, with the addition of metals from the IV-VI group of the periodic table of chemical elements, such as Nb [67], Ti [90,91], Mo [92], Ta [93], V [94], and W [95]. Research is also being conducted into the use of boron (B) systems. Mostly, ferrous alloys are used as starting components for hardfacing materials [64,65,68,90,92,93]; however, there are studies on the microstructure and properties of materials manufactured using pure starting components, such as metal powders [88,96,97] that have improved mechanical and tribological properties. For example, the results for systems with ferrous alloys (ferroalloys) as starting components [92] indicate that the addition of Mo leads to intensive cracking, while in systems with pure metal powders as starting components [97], cracking wasn′t observed with increasing amounts of Mo.

The analysis of recent scientific work on reinforcement technologies and the chemical composition of hardfacing electrodes shows that the development of new tungsten-free systems for flux-cored electrodes is a promising research area. As already mentioned, Fe-Cr-C and Fe-Ti-C, with the addition of some elements, are widely used systems for FCAW restoration and increasing the wear resistance of machine parts. However, there are also other systems that can be considered as prospects for the development of electrodes using pure metal powders instead of ferroalloys as starting components.

From this viewpoint Fe-rich alloys of the Fe-Mo-B-C alloying system is the promising candidates for hardfacing alloy development due to the presence of a hard and thermodynamically stable Fe(Mo,B)_2_ phase, known as τ_2_—phase [98], which can be formed during welding, as shown in [97,99,100]. Further improvement in the properties of such alloys can be achieved by refining the grain structure and increasing the properties of the steel matrix phase and expanding the alloy system. Therefore, the effects of the Ti and Mn additions on the microstructure, abrasion, and impact-abrasion wear resistance of the Fe-Mo-B-C hardfacing alloys were investigated.

Purpose and tasks of research.

The aim of this work was the design of the Fe-based hardfacing alloys within the Fe-Mo-Ti-B-C and Fe-Mo-Mn-B-C alloying systems for use as materials in FCAW hardfacing. To achieve this goal, the following tasks were set:(1)Determination of the optimal composition of the hardfacing alloys of the Fe-Mo-Ti-B-C and Fe-Mo-Mn-B-C systems in a range that is suitable for the production of flux-cored wires from pure metal powders as starting components;(2)Investigation of the effects of Ti and Mn additions on the microstructure and properties of Fe-Mo-B-C hardfacing alloys;(3)Determination of wear resistance by dry sand and rubber wheel abrasion and impact-abrasion tests compared to the wear resistance of commercially available hardfacing alloys.

## 2. Materials and Methods

### 2.1. Thermodynamic Analysis

The determination of the optimal composition of the hardfacing alloys of the Fe-Mo-Ti-B-C and Fe-Mo-Mn-B-C systems were performed using Thermo-Calc version 2022a software [101] with the Thermo-Calc Software TCFE12 Steels/Fe-alloys database [102]. The starting relationship between components for the thermodynamic calculations was chosen according to the possible amounts of the alloy components that can be placed into flux-cored wire with the steel sheath while maintaining the equimolar ratio between Mo and B.

### 2.2. Preparing Starting Components

In order to obtain a high intensity of chemical reactions and dissolution precipitation processes as well as high degree of alloying of the hardfacing layer, the commercially available metal powders of Ti (PTS-1 TU 14-22-57-92 grade manufactured by JSC “Titanium Institute”), Mo (MPCh TU 48-19-69-80 grade, manufactured by CJSC «Promelectronica»), and Mn of MN95 grade with the fine particle size (in range of 1–5 μm) were used. The B-containing compound powders were used as a technologically favorable source of B and C, which ensure the formation of the refractory hard phases. To protect the arc from the atmosphere and improve arc stability during welding, fluorite and rutile were added to the powder mixture. To dry the components before weighing, they were dried in a SNOL-type drying oven at a temperature of 120 °C for 1.2 h. Mixing of the components was carried out in a laboratory gravity tumble mixer with an inclined axis of rotation for 8 h. An additional drying at 120 °C for 0.5 h was performed to prevent the influence of air humidity on the quality of the powder mixture.

### 2.3. Manufacture of FCAW Electrodes

Experimental hardfacing materials were fabricated as an FCAW wire (according to ISO 14343:2017) with overlap joint (Figure 1) by drawing prepared flux powder into a sheath of 08kp low carbon steel (standard DSTU EN 10139:2018), with a size of 0.5 to 20 mm. Chemical composition and mechanical properties of 08kp steel are shown in Table 1 and Table 2. Such a construction of the FCAW wire contributes to more stable burning of the arc, as well as more uniform heat propagation through the flux charge. The experimental flux-cored wire was manufactured using the equipment of the Interdisciplinary Research and Production Center “Epsilon LTD”, Ivano-Frankivsk, Ukraine.

After the wire was made, it was cut into 420 mm long electrodes for convenient determination of electrode characteristics and application testing in manual arc mode. The determination of the chemical composition of the electrodes was based on the ratio (filling coefficient$\;K_{f}$) between powder filler and steel sheath
(1)$$K_{f}=100-\frac{W_{E}\cdot 100}{W_{O}}.$$
where $W_{E}$ is the weight of the filled electrode and $W_{O}$ is the weight of the empty electrode, which is equal to 28.5 g; the weighing accuracy was 0.01 g, and the calculation was performed by weighing three electrodes for each composition. The filling coefficient of the experimental electrodes was about 30%.

### 2.4. Hardfacing Process

A steel plate (material steel 40 standard DSTU 7809:2015) with a crosssection of 15 × 20 mm was chosen as the substrate. The chemical composition and mechanical properties of 08kp steel are shown in Table 3 and Table 4.

FCAW hardfacing mode was chosen to allow for a more intense transfer of the electrode material to the base material, with a direct current of 170 A with reversed polarity and an arc voltage of 30–32 V using a VDU-506 rectifier. Hardfacing coatings were manually deposited into three layers to minimize the effects of mixing the coating with the substrate material; thus, the coating thickness was about 5 mm. Air-cooling was carried out at 20 °C.

### 2.5. Microstructure Observation and Mechanical Properties Measurement

The microstructure of the hardfacing layer and the morphology of the worn surfaces was observed by scanning electron microscopy (SEM) using a ZEISS EVO 40XVP microscope at the Center for Collective Use of Scientific Instruments “Center for Electron Microscopy and X-ray Microanalysis” of the Karpenko Physico-Mechanical Institute of the National Academy of Sciences of Ukraine, Lviv, Ukraine. The chemical composition of the phases was examined using energy-dispersive X-ray spectroscopy (EDS). Considering the difficulties in identifying boron and carbon using the EDS technique, the composition of boride phases was determined based on the ratio of metal components, and the carbon content was determined only for carbide phases. The microhardness distribution at the interface between the hardfacing layer and the base metal was measured with the Vickers pyramid at a load of 0.1 N, using a PMT-3 hardness tester. Macro-hardness was measured by means of the average measurements taken from the top surface of the experimental hardfacing coatings and using the Rockwell method, scale “C”, with a modernized TK-2 hardness tester.

### 2.6. Abrasive Wear Test in Loose Abrasive Condition

Tribological tests of hardfacing alloys were performed in the loose abrasive state, since such a wear mechanism is most common for machine parts in the metallurgical, agricultural, woodworking, and mining industries. Such a wear mechanism is also one of the main causes of wear during the operation of machine parts under conditions of intense abrasive action. The abrasive wear tests were performed on the developed testing machine of material in a loose abrasive condition (Figure 2), which was designed with regard to the recommendations of the standard (ASTM G65-16 (2021) Standard Test Method for Measuring Abrasion Using the Dry Sand/Rubber Wheel Apparatus). The test process is as follows:

The fixing lever (1), installed on a hinged support, presses the hardfacing flat sample (10) to the rubber wheel (5). The required value of pressing force is provided with the appropriate selection of the value of *L_i_*, installation of the removable weight (2), and selection of its mass. The supply of abrasive particles (9) flows through the bowl (4) into the friction zone formed by contact of the flat sample (10) with the rubber wheel’s cylindrical surface (5). The linear velocity vector of the rubber wheel (5) coincides with the direction of flow of the abrasive particles (9) into the friction zone. Due to the friction between the rubber wheel’s cylindrical surface (5) and the surface of the flat sample (10), abrasive particles (9) are thrown into the friction zone, partially deepened into the rubber, and then interact with the hardfacing, causing wear due to scratching and cyclic deformation.

The properties of the rubber wheel are as follows: diameter, *D* = 50 mm; width, *b* = (15 ± 0.1) mm; hardness of the rubber, 78–85 Shore units; relative residual elongation of the roller material at break, 15–20%. The harness of the rubber surfaces was measured according to the ASTM 2240 Standard using a tire durometer of type “D”. The proposed installation of the fixing lever at an angle to the horizontal creates improved conditions for the supply of abrasive particles into the friction zone. The non-parallelism of the horizontal axis of the rubber wheel to the working surface of the flat sample was not more than 0.1 mm.

The testing machine (Figure 3) consists of an electric motor (6) equipped with a device for regulating the speed of the motor (7), causing the rubber wheel (5) to rotate about a horizontal axis.

The wear tests were performed using a silica “Alfa-Quartz” (SiO_2_) sand abrasive manufactured by Ekkom Plus, Kyiv, Ukraine with a grain size in range of 0.2–0.4 mm. The content of impurities in abrasive mass was not more than 2% (Standard DSTU B V.2.7-131:2007). As can be seen (Figure 4), particles are characterized by a non-equiaxed grain shape with sharp edges at the vertices, indicating their high abrasive ability.

Samples with the experimental hardfacing layer and standard materials were made in the form of samples measuring 15 mm wide, 20 mm long, and 25 mm high, with dimensional tolerances according to the 14th quality of accuracy and roughness and a work surface not lower than 1.25 μm.

The friction mode for wear resistance tests was chosen as follows: load, *p* = 2.4 kN; roller speed converted to linear speed, 30 m/min; test duration, 220 min. The duration of the tribological tests was measured with a stopwatch.

The amount of weight loss due to wear was determined by gravimetric method: weighing the test specimens on the analytical axis of VLA-200 g-M (weight accuracy of 0.1 mg) before and after the test. The weight loss of the sample due to wear during the tests was at least 5 mg. Before each weighing, the samples were thoroughly wiped with alcohol and dried. The arithmetic mean value from three tests for wear of the hardfacing layer was chosen as the resulting value for determining the wear resistance.

The impact-abrasion wear tests were conducted using a laboratory machine that provided the mode in which a cemented carbide striker impacts the sample with an abrasive environment present in the contact zone. The impact-abrasion wear tests were conducted using a laboratory machine described in [103] providing the mode in which a cemented W-carbide striker impacts the sample with an abrasive environment present in the contact zone. The impact energy was set to 2.4 J/cm^2^ and the chilled white cast iron particles with an average size of 1 mm were chosen as the abrasive environment.

## 3. Results and Discussion

### 3.1. Thermodynamic Modelling

In the case of replacing Mo in the base alloy with Ti in the range of 0.16 mol %, the polythermal section representing the phase equilibrium in the Fe-Mo-Ti-B-C system is shown in Figure 5. As can be seen from the figure, in the high-temperature regions and when the concentration of Ti exceeds 2 mol %, crystallization starts from the formation of TiC with NaCl structure. This compound can act as the nucleation site for the following formation of the Fe(Mo,B)_2_. At the regions below the solidus line, the increasing Ti content in the range of 1.2–4.75 mol % causes the stabilization of the Fe_2_B phase formed by a peritectoid-type reaction. In the regions below the solidus line, the increase in Ti content in the range of 1.2–4.75 mol % causes the stabilization of the Fe_2_B phase, which is formed by a peritectoid-type reaction. The further increase in Ti content above 4.75 leads to the formation of complex (Ti,Mo)B boride phase, which partially replaces the Fe(Mo,B)_2_. Considering the necessity of bonding all of the C into TiC, preventing the formation of the cementite and avoiding the formation of unfavorable (Ti,Mo)B phase the optimal Ti concentration is in the range of 4.1–4.6 mol % (the corresponding equilibrium region marked with green color in Figure 5a). So, the composition in the experimental hardfacing coating corresponding to Sample 2 was set to the 4.3 mol %.

The calculated polythermal section of the Fe-Mo-Mn-B-C system corresponding to hardfacing alloys with different Mn content (balance Fe) is shown in Figure 5b. As can be seen in Figure 5b, the increase in Mn content in the range of 0–20 mol % leads to the decrease of melting temperatures and significant stabilization of the austenite phase. Crystallization of the alloys within the investigated compositional range begins from the formation of the Fe(Mo,B)_2_, followed by the formation of the austenite phase. In the solid state, Mn-austenite coexists with the Fe(Mo,B)_2_ and other phases over a very wide range of temperatures and concentrations. The formation of undesirable phases, such as (Fe,Mn)_2_B-type complex boride and M_7_C_3_, M_2_C-type carbides containing significant amounts of Mn, are formed when the Mn content exceeds 6 mol %, while the ξ-carbide forms in the solid state when the Mn content is in the range of 0–4 mol %. Considering this limitation, and with the aim of providing maximum possible austenite stability within the Austenite + (Fe,Mn)_2_B + Cementite three-phase region (marked with green color in Figure 5b), the composition of Sample 3 was adjusted to meet the 6 mol % of Mn in the system.

The temperature dependencies of the phase composition during solidification for the alloys corresponding to the experimental hardfacing alloys are shown in Figure 6a–c. The solidification of the base alloy (Sample 1) starts at a temperature of about 1750 K (Figure 6a) from the crystallization of the Fe(Mo,B)_2_ phase, and after increasing its amount up to 0.3 volume fraction at 1500 K, eutectic decomposition of the liquid into a mixture of austenite and Fe(Mo,B)_2_ takes place, and further cooling in the range of 1500 to 1000 K leads to the precipitation of the cementite phase. After the eutectoid reaction of the austenite decomposition into Ferrite and Cemenite occurs, the resulting predicted structure consists of primary Fe(Mo,B)_2_, eutectic Fe(Mo,B)_2_ + Austenite, and the Pearlite-like eutectoid. The solidification process of Sample 3 is very similar to the base alloy (Sample 1) except for the broader temperature ranges of eutectic and eutectoid reactions and lower cementite stability (Figure 6c). The solidification of the alloy, according to Sample 3, starts with the formation of the Ti carbide at a temperature of about 1800 K, then the Fe(Mo,B)_2_ phase crystallizes, and the amount of both phases increases simultaneously during cooling. At 1500 K, the eutectic (Fe(Mo,B)_2_ + Austenite) reaction occurs and the complex eutectoid-like reaction including Fe(Mo,B)_2_, Austenite, Fe_2_B, and TiC takes place at 1250 K. The resulting structure thus consists of four phases (Ferrite, Fe(Mo,B)_2_, Fe_2_B, and TiC), forming primary phases (carbide and boride), eutectic and eutectoid. The main differences between the phase composition of the alloy solidified under real conditions and the equilibrium state are characteristic of austenitic systems with high levels of dissolved Mn, similar to the composition of Sample 3. This is caused by a strong austenite-stabilizing effect of Mn, which prevents eutectoid decomposition of austenite into a ferrite–cementite mixture. Thus, the structure of the real alloys cast under normal conditions corresponds to high-temperature regions (above 1000 K) of the phase diagram and the equilibrium structure can be obtained through the long annealing that provides high intensity of the diffusion processes. For the alloy studied, rapid cooling conditions during arc welding allow the high-temperature phase region (filled in green in Figure 5b) to be fixed and, as a result, a stable work-hardenable Mn-austenite is obtained at room temperature.

For the manufacturing of the experimental flux-cored wires the calculated compositions of the Samples 1–3 were converted from mol % into wt % and compositions were adjusted by filling coefficients to obtain the values, which are listed in Table 5.

### 3.2. Structure

The EDS maps of the hardfacing layers and the results of the EDS analysis in local areas are shown in Figure 7 and Figure 8 (Table 6), respectively.

As can be seen in Figure 7a, the Mo is mainly concentrated in the areas corresponding to the inclusions of the faceted grains (white phase), while other areas are mainly enriched with Fe. Results of local EDS analysis (Figure 8a, Table 6) at the white grains show that the relationship between metal components (Fe and Mo) is close to the single-point thermodynamic calculation results, which is shown in Figure 6a and corresponds to the stochiometric relationship in the FeMo_2_B_2_ compound. Sample 2 (Figure 7b) shows similar Mo enrichment in areas corresponding to the FeMo_2_B_2_ and surrounding colonies with eutectic morphology, while the Ti is mainly concentrated within the FeMo_2_B_2_ grains, indicating its modification activity. The chemical composition of the central (dark) regions of the FeMo_2_B_2_ grains, according to local EDS scanning (Figure 8b, Table 6), can be approximated by the carbide phase with the formula unit Ti_0.4_Mo_0.1_C_0.5_. The results of the equilibrium calculation (Figure 6b) also show the presence of a TiC-based carbide phase, but the amount of dissolved Mo is significantly lower. The EDS map of Sample 3 (Figure 7c) shows the preferred location of Mo within the areas corresponding to the white faceted FeMo_2_B_2_ grains and the areas with multiphase lamellar structure, while Fe and Mn are relatively evenly distributed outside the Mo-enriched regions. According to local EDS analysis (Figure 8c, Table 6), the lamellar structure has significant amounts of Fe, Mn, and Mo, indicating the presence of different phases in the analyzed area. The gray Fe-based phase with dendritic structure has large amounts of Mn (approximately 10 wt %), which agrees well with the equilibrium single-point calculation results (Figure 6c). Such a high Mn concentration makes it possible to stabilize the austenite phase with work-hardening ability.

In order to obtain microstructure observation results suitable for the comparative study, the analysis was performed on the areas equidistant (200 µm) from the visible interface between the deposit and the base steel. The microstructure of Sample 1 (Figure 9a) comprises faceted grains with an irregular shape and an average size of 10 µm (white phase), evenly distributed throughout the structure.

According to the EDS analysis, this phase was identified as a Fe(Mo,B)_2_. The Fe(Mo,B)_2_ grains are surrounded by eutectics with a Chinese Script (CS) morphology consisting of the mixture of the Fe(Mo,B)_2_ and Ferrite. The remaining area in the inspection plane is filled with Ferrite (dark gray phase). The results of the microstructure observations of Sample 2 (Figure 9b) show that Ti additions in a given amount have a strong effect on the structure formation processes. The dispersed inclusions of primary TiC (dark phase) reside in the central regions of the Fe(Mo,B)_2_ grains, providing significant grain refinement (average grain size is close to 5 microns) and a much more equiaxed grain shape. The remainder at the analysis plane is mainly occupied by complex eutectoids composed of at least four phases (Fe_2_B, Fe(Mo,B)_2_, Ferrite and TiC) and small areas of Ferrite (dark gray phase) distributed around the Fe(Mo,B)_2_ grains. In the case of alloying with Mn (Sample 3), structure changes mainly concern the formation of large dendritic grains (dark gray phase) growing normally in the interface between the hardfacing layer and the base steel (Figure 9c). The interdendritic space is filled with fine lamellar eutectics (Fe(Mo,B)_2_ + Austenite) and the fine inclusions of the primary Fe(Mo,B)_2_ with the acicular shape.

A comparison of the results calculated in the CALPHAD approach (Figure 5 and Figure 6) with the experimental data obtained by SEM and EDS methods (Figure 7, Figure 8 and Figure 9 and Table 6) shows that the thermodynamic modeling allows a prediction of stability for regions and amounts of all key phases that provide wear resistance (FeMo_2_B_2_, Ferrite, Austenite and TiC) and eutectic reactions involving such phases. However, cementite and carbides of the M_7_C_3_ and MoC types tend to be unstable under alloy structure formation conditions during hardfacing. This can be caused by their decomposition due to arcing and significant C burnout. The calculated chemical composition of the phases and the experimental data also agree well, but the exceptions are for the solubility of Mo in TiC and Mn in FeMo_2_B_2_. In both cases, the experimental values are significantly higher than the calculated ones. In the case of TiC, the formation of complex (Ti,Mo)C carbide is thermodynamically favorable at the high temperatures, but the equilibrium solubility of Mo is negligible under standard conditions. In real systems, the (Ti,Mo)C carbides form around the primary TiC grains at high temperatures in the form of a core-rim structure that remains stable down to low temperatures. Achieving the equilibrium microstructure under such conditions is possible by diffusion annealing, but considering the hardness of (Ti,Mo)C, which is higher than that of pure TiC, this operation is not required for wear-resistant coatings. The calculated solubility of Mn in FeMo_2_B_2_ is equal to 0, while the results of experimental investigations show clear traces of Mn over the FeMo_2_B_2_ grains. Such differences are generally caused by the lack of thermodynamic data for pure MnMo_2_B_2_, as well as excess terms representing the mixing energy between FeMo_2_B_2_ and MnMo_2_B_2_, which are used to evaluate the free energy of the (Fe,Mn)Mo_2_B_2_ phase within the CEF (compound energy formalism) model for the phase consisting of three sublattices filled with Mo, Fe + Mn, and B.

### 3.3. Properties

In order to compare the relative wear resistance of the experimental hardfacing alloys to the known materials, heat-treated grade R6M5 high-speed steel (HSS) was selected as the standard specimen for comparative wear resistance studies. The results of the comparison of the hardness and wear resistance are shown in Figure 10. As can be seen, the hardness of all experimental hardfacing alloys exceeds 60 HRC. The Fe-Mo-Ti-B-C system alloy corresponding to Sample 2 has the highest abrasion resistance, as well as a hardness value (65 HRC) caused by the bonding of C in the TiC compound, with a high (30 GPa) microhardness value and the presence of large amounts of eutectoids comprising hard boride phases (Fe_2_B and Fe(Mo,B)_2_). Despite this, the impact wear resistance of this alloy is relatively low compared to Samples 1 and 2, due to the presence of a continuous eutectoid network with low crack resistance and small amounts of high plasticity ferrite located in closed volumes. In contrast, the addition of Mn to the basic Fe-Mo-B-C system led to a decrease in abrasion resistance and hardness values (from 64 to 62 HRC), while impact wear resistance increased significantly. Such property changes are primarily caused by the formation of the large amounts of Mn-austenite that form disclosed volumes. The hardness of Mn-austenite in the initial undeformed condition (250 HB) does not allow effective resistance to abrasive wear cause by the micro-cutting mechanism, but in the case of work hardening during impact loading, the micro-hardness of Mn-austenite increases by about two times, allowing it to obtain sufficient resistance to the penetration of abrasive particles.

For comparison, we performed wear tests for chrome-coated samples, according to the technology [42]. The test results show that their wear resistance was at the level of high-speed steel but lower than the welded samples.

The micrographs of the worn surfaces are shown in Figure 11. As can be seen, the morphology of the worn surface of Sample 1 (Figure 11a) of this alloy shows strong binding of the hard Fe(Mo,B)_2_ inclusions with the matrix, but the presence of deep scratches without preferred orientation indicate that the abrasive particles penetrate the material, providing a micro-cutting process and “bypassing” hard grains in the matrix. Increased wear resistance of Sample 2 is observed wherein the matrix exhibits higher wear resistance with abrasive particles, and hard grains in the material serve as barriers to abrasive particle penetration. Additionally, the nature of the wear indicates that the wear process differs from the mechanisms discussed above. Traces of scratches (Figure 11b) are isolated and their depth is insignificant, suggesting that the wear is not due to material separation in the form of chips or cracking pieces of material.

Analysis of the worn surface of Sample 3 (Figure 11c) shows that wear occurs through multiple mechanisms. Scratches that occur while the abrasive particles penetrates the hardfacing, followed by their movement in the material, can be seen on the worn surface. The result of this action is plastic deformation in the cutting area and separation of the material in the form of chips, depending on the penetration angle of the abrasive grain into the material [104]. Concentrated depressions of relatively regular shape are also visible on the relief, which can be the result of the detachment of the hard grains of the material under the action of the abrasive, occurring when the solid inclusion protrudes from the matrix and the connection between them are not strong enough or the matrix is not hard enough to hold the grain.

The analysis of the worn surfaces after the impact-abrasion wear tests (Figure 10) shows that the wear of the HSS sample (Figure 12a) is generally caused by micro-fatigue wear, which arises as a result of the connection of multiple fatigue cracks into local closed networks, causing the delamination of large volumes of material. Despite the high hardness of the material, the surface is riddled with scratches that can serve as nucleation sites for the formation of fatigue cracks that can continue to grow. In contrast, the surface of the Mn-alloyed Sample 3 (Figure 12b) has the relatively smooth relief covered by small volumes with the traces of intense plastic deformation. The inclusions of the Fe(Mo,B)_2_ remains embedded in the steel matrix preventing formation of deep scratches. The resulting wear thus occurs as a result of the detachment of the relatively small volumes of deformed material.

According to the conducted investigations, the structure of the developed tungsten-free hardfacing alloys with pure metal powders as the starting components differs from the structure of alloys with a similar chemical composition, but obtained with ferrous alloys as the starting components, in which the increase in the Mo content led to cracking [92]. Furthermore, in the above studies, the use of Mo as a ferroalloy does not lead to the formation of Fe(Mo,B)_2_, which was observed in all experimental alloys.

However, in the work [98], the B-Fe-Mo system was researched, where experimental samples were made in a vacuum furnace using a non-consumable tungsten electrode under pure argon gas with pure Mo. In the work, a Fe(Mo,B)_2_ phase was observed, similar to the phase in the microstructure of the experimental alloys, which were deposited by FCAW.

Upon observation of the experimental hardfacing alloy’s microstructure, the structure of Sample 2 is characterized by the absence of the TiB_2_ phase. However, there are works [46,104] that indicate the possibility of formation of titanium carbides and titanium diborides in the structure, which is not observed in the structure of this sample. Such differences can be the result of the presence of Mo in the chemical composition of the electrode, as well as the use of different brands of starting components.

The wear test results show that the wear resistance of the HSS sample is significantly lower compared to the experimental hardfacing alloys, especially when they contain both a high TiC and Fe(Mo,B)_2_ content. This is due both to the high microhardness of the phases and to their distribution in the structure, which is more favorable because it corresponds to the matrix-reinforced structure characteristic of composite materials and tungsten-cemented carbides.

The extension of the Fe-Mo-B-C alloy system by Mn addition makes it possible to obtain Mn-austenite similar to the austenite of a Hadfield steel [105] with work-hardening ability due to the twinning-induced plasticity. In the hardfacing alloy corresponding to Sample 3, such Mn-austenite coexists with the hard Fe(Mo,B)_2_ phase, allowing for effective protection against the impact-abrasion wear.

## 4. Conclusions

The optimal composition of the Fe-rich alloys of Fe-Mo-Ti-B-C and Fe-Mo-Mn-B-C was determined using the CALPHAD approach as implemented in the Thermo-Calc software. Comparing the modeling results with experiments agrees well with the compositional ranges that are promising for the development of hardfacing alloys.

It has been observed that the addition of Ti to the Fe-Mo-B-C hardfacing alloy results in significant grain refinement due to the formation of TiC, which acts as a structural modifier, while the Mn addition results in the formation of Mn-austenite in the shape of dendritic grains and lamellar eutectic Mn-austenite and Fe(Mo,B)_2_.

The abrasion and impact-abrasion wear tests show that alloying the Fe-Mo-B-C system hardfacing alloys with Ti results in an increase of up to 1.2 times in their abrasion wear resistance, and the addition of Mn results in an increase in impact-abrasion resistance up to 1.3 times. That creates prerequisites for the industrial use of developed alloys in the form of flux-cored wires for FCAW hardfacing.

Further investigations need to be carried out on the effects of Mn dissolution on the mechanical properties and thermodynamic stability of Fe(Mo,B)_2_.

## Figures and Tables

**Figure 1 materials-15-05074-f001:**
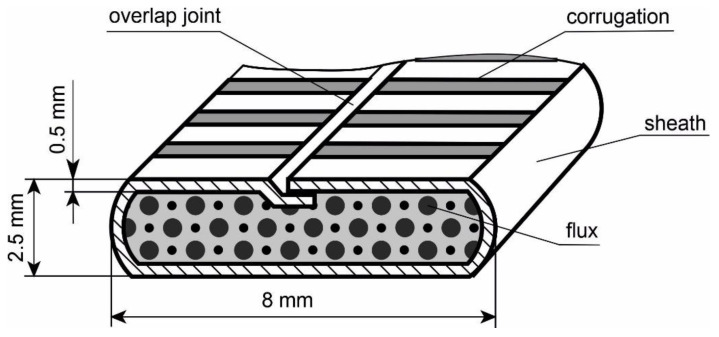
Schematic representation of the cross section of an FCAW wire.

**Figure 2 materials-15-05074-f002:**
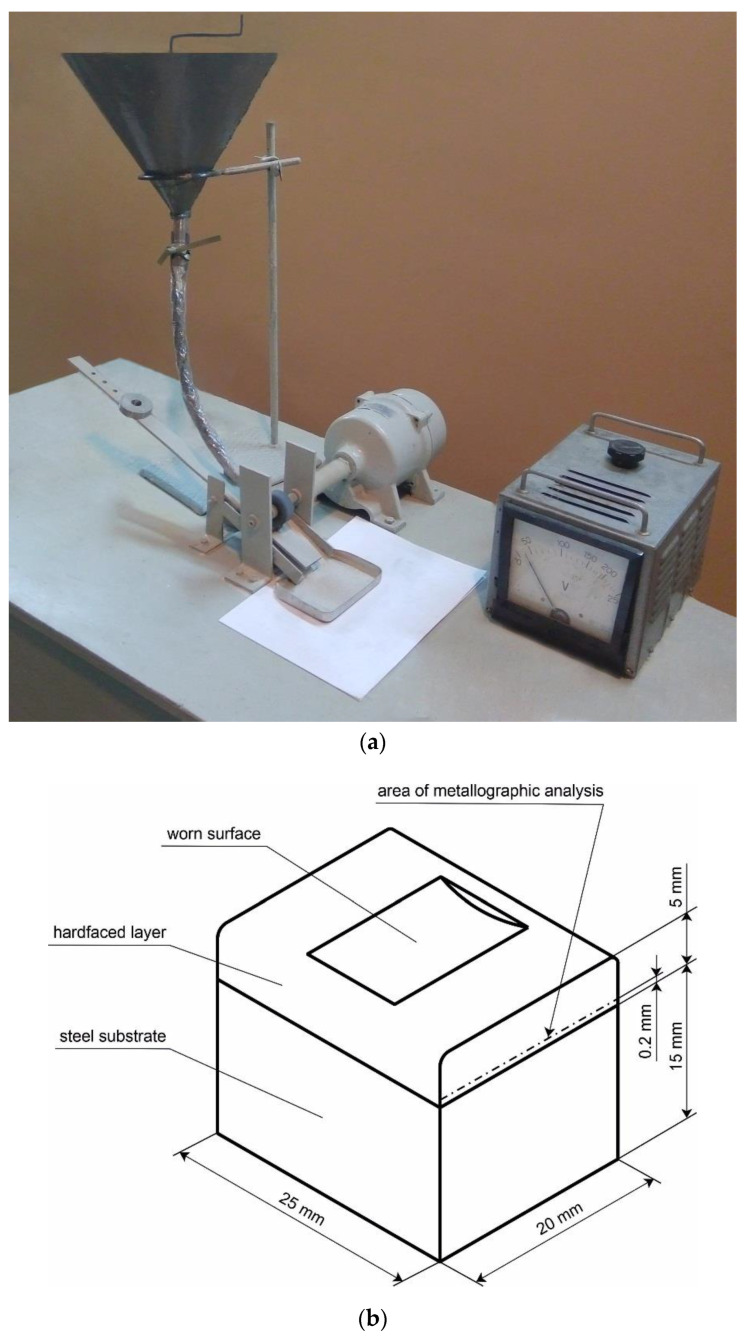
General view of the dry sand and rubber wheel abrasion testing machine (**a**) and flat sample with hardfacing layer (**b**).

**Figure 3 materials-15-05074-f003:**
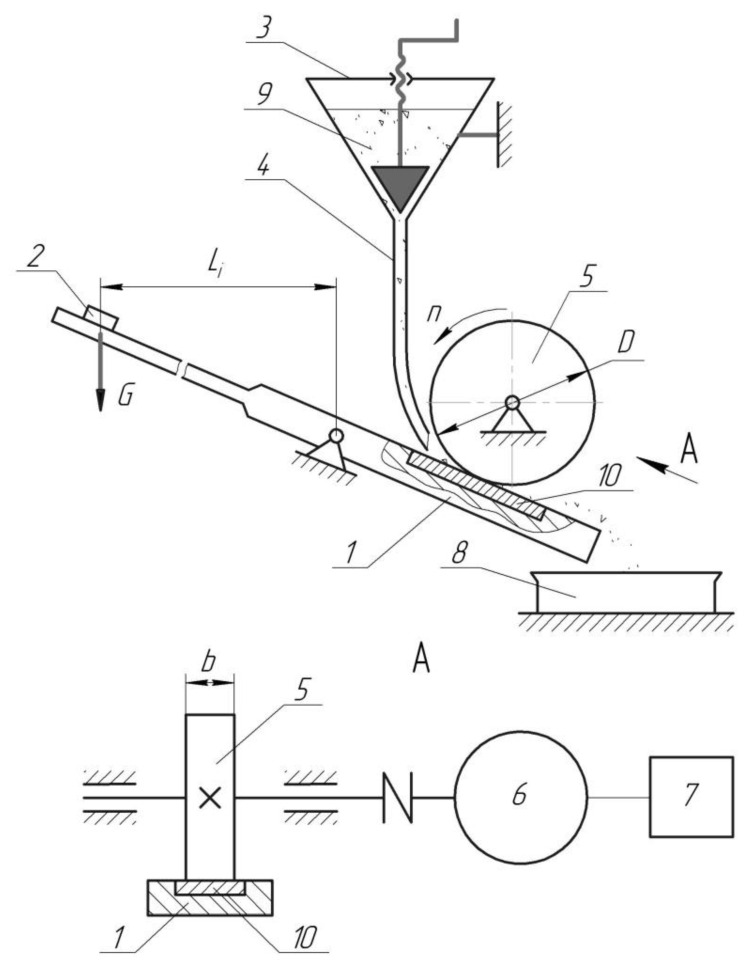
Scheme of the dry sand and rubber wheel abrasion testing machine: 1—sample-fixing lever, 2—removable weight, 3—abrasive hopper, 4—nozzle for supplying of an abrasive particles into the friction zone, 5—rubber wheel, 6—electric motor, 7—device for regulating the speed of the engine, 8—container for used abrasive, 9—abrasive particles, 10—flat sample with hardfacing layer.

**Figure 4 materials-15-05074-f004:**
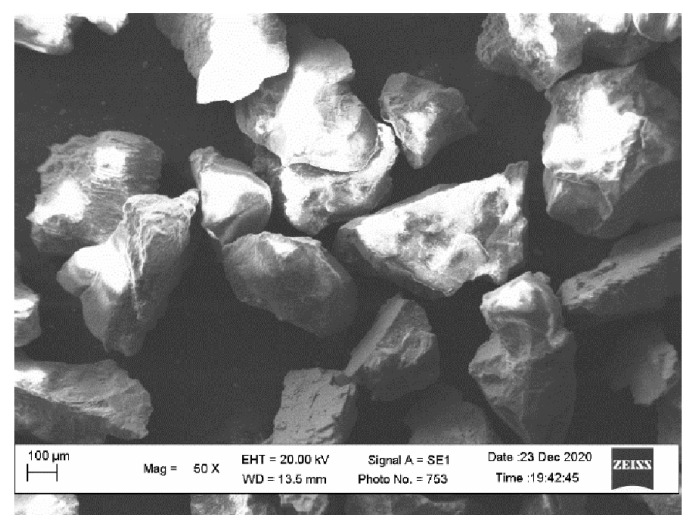
The results of the studies of the morphology of the SiO_2_ abrasive particles.

**Figure 5 materials-15-05074-f005:**
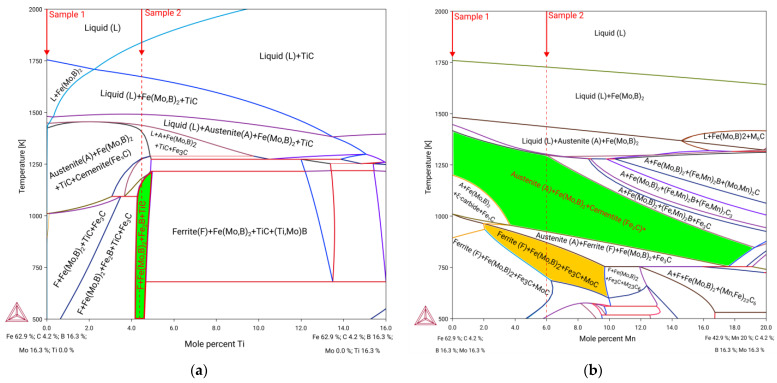
The polythermal sections of the Fe-Mo-Ti-B-C (**a**) and Fe-Mo-Ti-B-C (**b**) systems.

**Figure 6 materials-15-05074-f006:**
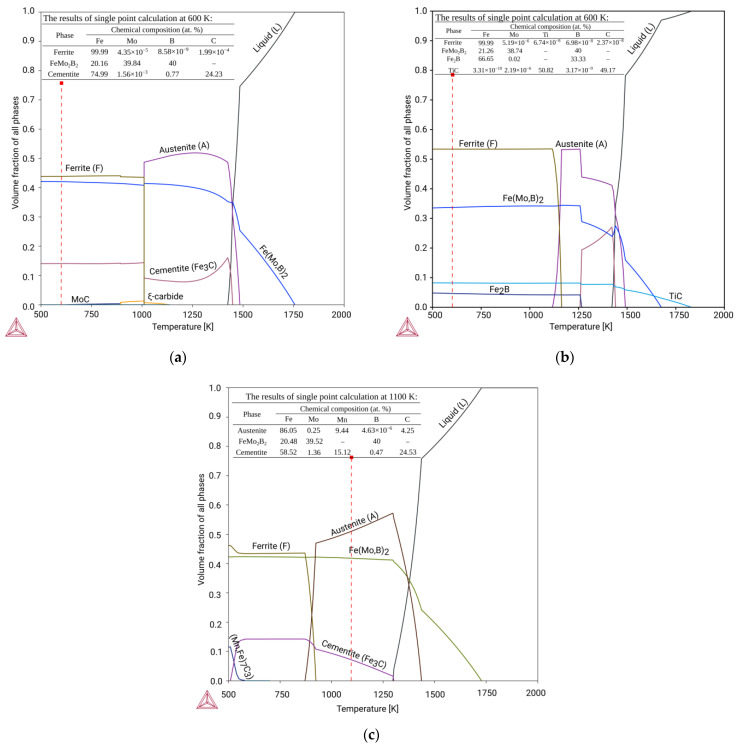
Calculated temperature step diagrams of experimental hardfacing alloys: (**a**) Sample 1; (**b**) Sample 2; (**c**) Sample 3.

**Figure 7 materials-15-05074-f007:**
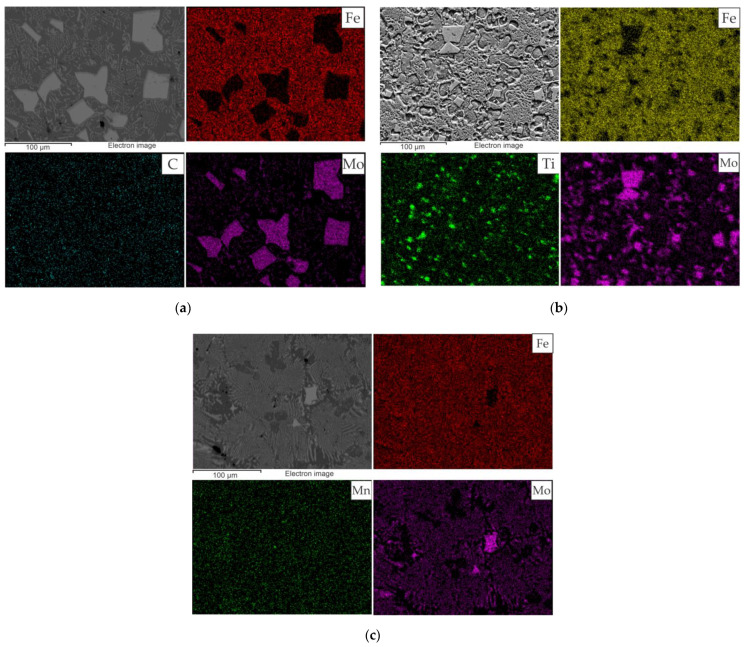
The EDS maps of the hardfacing layers: (**a**) Sample 1; (**b**) Sample 2; (**c**) Sample 3.

**Figure 8 materials-15-05074-f008:**
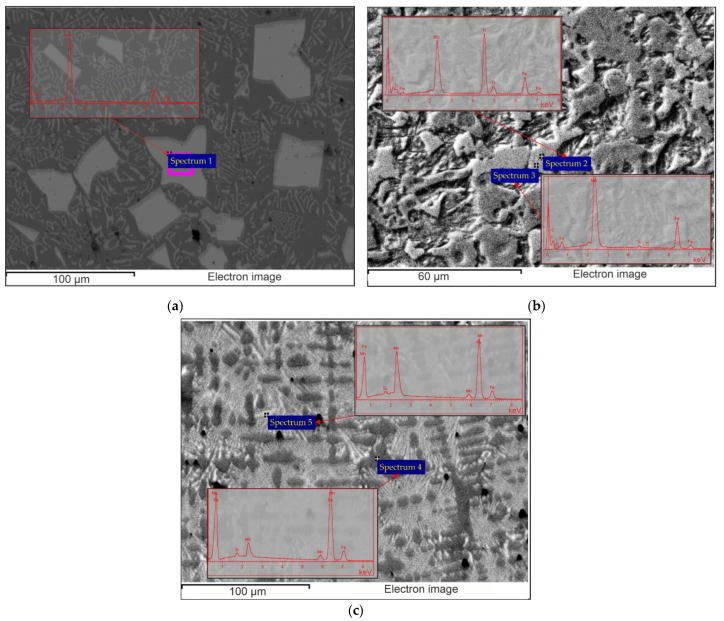
The local areas of the EDS analysis and the corresponding spectra: (**a**) Sample 1; (**b**) Sample 2; (**c**) Sample 3.

**Figure 9 materials-15-05074-f009:**
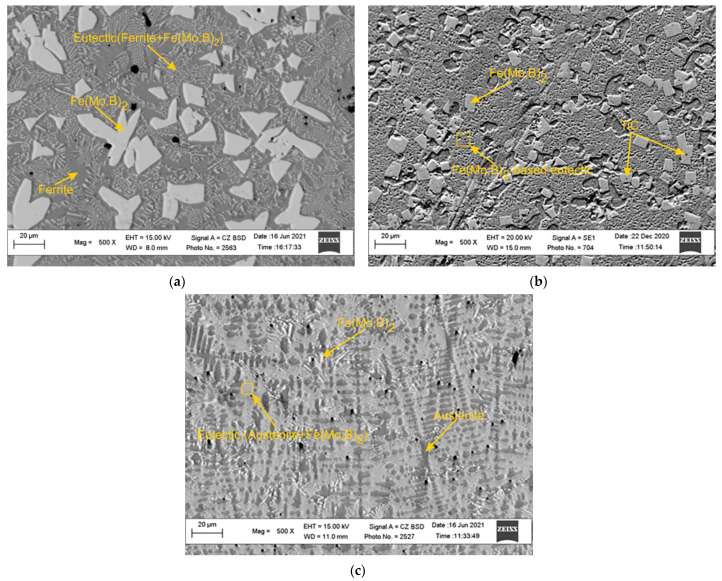
The microstructure of the experimental hardfacing alloys: (**a**) Sample 1; (**b**) Sample 2; (**c**) Sample 3.

**Figure 10 materials-15-05074-f010:**
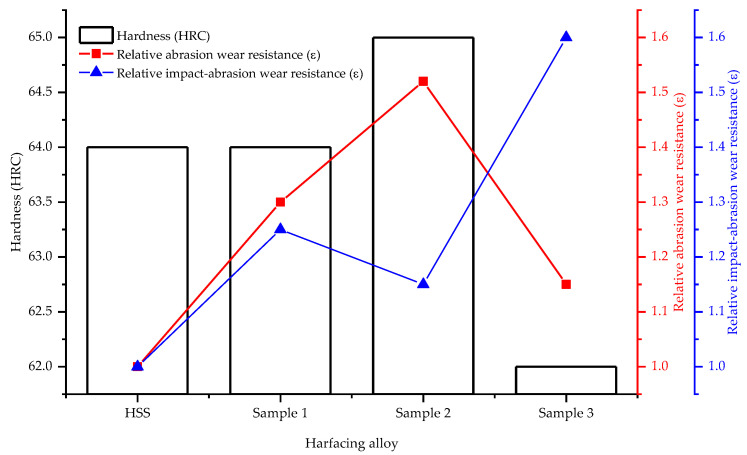
Comparison chart of hardness and wear resistance of experimental hardfacing alloys.

**Figure 11 materials-15-05074-f011:**
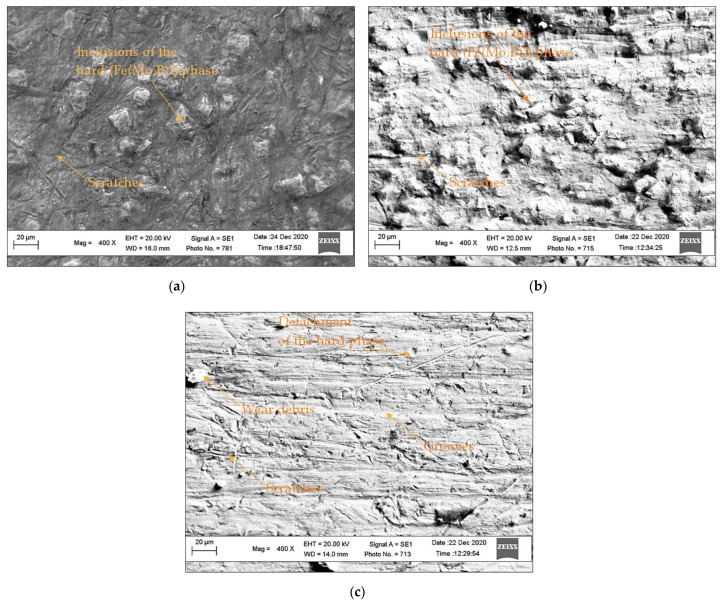
Morphology of the worn surfaces of hardfacing alloys after abrasion tests: (**a**) Sample 1; (**b**) Sample 2; (**c**) Sample 3.

**Figure 12 materials-15-05074-f012:**
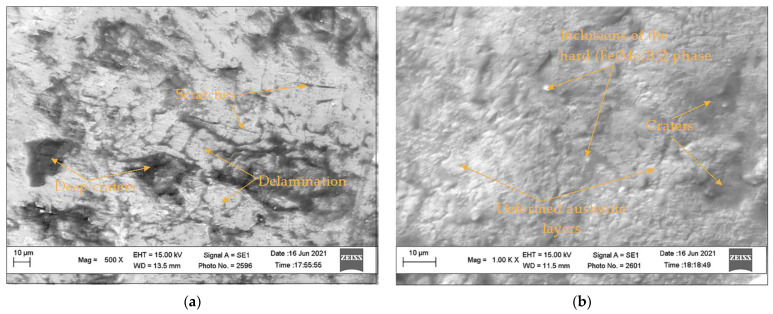
The SEM micrographs of the worn surfaces after impact-abrasion wear tests: (**a**) HSS sample; (**b**) Sample 3.

**Table 1 materials-15-05074-t001:** Chemical composition of 08kp steel, wt %.

C	Si	Mn	Ni	S	P	Cr	Cu	As
0.05–0.11	<0.03	0.25–0.5	<0.25	<0.04	<0.035	<0.1	<0.25	<0.08

**Table 2 materials-15-05074-t002:** Mechanical properties of the 08kp steel.

Hardness, MPa	Short-Term Strength Limit, Mpa	Ultimate Elongation, %	Density (at 20 °C), kg/m^3^
131	290	60	7871

**Table 3 materials-15-05074-t003:** Chemical composition of steel 40, wt %.

C	Si	Mn	Ni	S	P	Cr	Cu	As
0.37–0.45	0.17–0.37	0.5–0.8	<0.3	<0.04	<0.035	<0.25	<0.3	<0.08

**Table 4 materials-15-05074-t004:** Mechanical properties of steel 40.

Hardness, MPa	Short-Term Strength Limit, MPa	Ultimate Elongation, %	Density (at 20 °C), kg/m^3^
241	610	35	7850

**Table 5 materials-15-05074-t005:** Chemical composition of experimental electrodes.

Sample	Chemical Composition (wt %)					
	Ti	Mn	Mo	B	C	Fe
Sample 1	-	-	29.5	3.5	1.0	66.0
Sample 2	4.0	-	22.5	3.5	1.0	69.0
Sample 3	-	6.0	30.0	3.5	1.0	59.5

**Table 6 materials-15-05074-t006:** Results of the EDS analysis at the local areas from Figure 9.

Sample	Spectrum	Chemical Composition (wt %)					
		Fe	Mo	Mn	Ti	C	Si
Sample 1	Spectrum 1	34.69	65.31	-	-	-	-
Sample 2	Spectrum 2	1.91	11.28	-	34.98	51.83	-
	Spectrum 3	-	-	-	-	-	-
Sample 3	Spectrum 4	85.29	2.74	10.29	-	-	1.68
	Spectrum 5	78.44	15.96	4.15	-	-	1.45

## Data Availability

Not applicable.

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
