# Peer review of "Microstructure and Wear Characterization of the Fe-Mo-B-C—Based Hardfacing Alloys Deposited by Flux-Cored Arc Welding"

_materials, 2022, doi:10.3390/ma15145074_

Round 1
Reviewer 1 Report
The research content and results of this paper have theoretical significance and engineering value. Please make the necessary revise or supplementary to the following questions.
1. At line 184, the Tab4 and Tab5 changes to the Tab3 and Tab4.
2. The use of EDS analysis was mentioned in the Materials and methods, but no results were seen in the main text. Wear-resistant layer thickness by depositing should be given.
3. The sentences expressed in lines 291-293 are repeated in lines 287-289291-293.
4. The various phases annotated in Figure 7 were determined by EDS analysis and are suggested to supplement the EDS analysis results.
5. The expression mode of chemical elements in the text should be unified. That is, they all use the chemical element symbol or the element name words.
6. The content of '4. Discussion' is too simple that it is not necessary to form separate sections. It is recommended to merge with '3.Results'.
Author Response
Dear Reviewer!
We sent the responses to the remarks in the attached file.
Best regards,
Authors

Reviewer 2 Report
The authors have conducted an study into the wear behavior of alloys prepared within the Fe-Mo-B-C system. While the research warrants publication the manuscript in the current for needs significant revision before being considered for publication.
Overall the English expression could be improved. Particularly the use of the word "technology" seems to be over used, often times leaving the true meaning or context ambiguous to the reader.
In the abstract line 30 I think it should say Ti and Mn (not Mo)
line 75 should be emergencies
line 288 is missing a figure number
line 359 etalon I am not sure if this is meant to be something like "baseline"?
Overall the introduction could use revising to tell a single cohesive story of the motivation behind the work, currently the intro goes back and forth too much, most notably from he inconsistent introduction and use of acronyms
Overall the results section could be rewritten in a more concise manner, making more consistent the way in which samples are described.
The description of thermodynamic calculations are a little confusing, and the language used sounds like the description of solidification but it is important to emphasize the difference between the equilibrium calculations and what occurs during solidification. A common term for the plots of fig 6 are "step diagrams".
Fig5 the two isopleths should have the same temp range on the y axis.
The differences between the step diagrams and the isopleths should be explained. when comparing particularly sample 1 the step diagram show different phases (ferrite instead of FeMoB2).
The arrow types in fig7 are varied and confusing.
It may help the reader if the location of the micrograph with respect to the weld is drawn schematically.
In sample three it is not clear which phase the dendrites are. Clarity and consistence in the reporting of phases by gray scale from the sem micrographs would help. EDS results reported would improve the clarity. Can the composition of each phase from EDS be compared to the composition from CALPHAD single point calculations predicting phase composition?
The discussion is lacking, it read like a conclusion section. More thoughtful discussion comparing the CALPHAD to the experimentally observed microstructures is needed.
Author Response

(The authors gave the same response as above.)

Round 2
Reviewer 1 Report
No!
Author Response
Dear Reviewer!
Once again, thank you very much for taking the time, and valuable comments that helped improve our article.
Best regards,
On behalf of the authors
Michał Bembenek
Reviewer 2 Report
The authors did well to address the comments made. The only issue with the draft document I received was unnecessary "-" in some words left from editing maybe.
Author Response
Dear Reviewer!
Once again, thank you very much for taking the time, and valuable comments that helped improve our article. We checked and corrected all the mistakes in the article.
Best regards,
On behalf of the authors
Michał Bembenek